# FedPop: A Bayesian Approach for Personalised Federated Learning

**Nikita Kotelevskii**[*]
Skolkovo Institute of Science and Technology
Moscow, Russia
Nikita.Kotelevskii@skoltech.ru
[*] Equal contribution

**Maxime Vono**[*]
Criteo AI Lab
Paris, France
m.vono@criteo.com
[*] Equal contribution

**Alain Durmus**
ENS Paris-Saclay
alain.durmus@ens-paris-saclay.fr

**Eric Moulines**
Ecole Polytechnique
eric.moulines@polytechnique.edu

## Abstract

Personalised federated learning (FL) aims at collaboratively learning a machine learning model tailored for each client. Albeit promising advances have been made in this direction, most of existing approaches do not allow for uncertainty quantification which is crucial in many applications. In addition, personalisation in the cross-silo and cross-device setting still involves important issues, especially for new clients or those having small number of observations. This paper aims at filling these gaps. To this end, we propose a novel methodology coined `FedPop` by recasting personalised FL into the population modeling paradigm where clients' models involve *fixed* common population parameters and *random* effects, aiming at explaining data heterogeneity. To derive convergence guarantees for our scheme, we introduce a new class of federated stochastic optimisation algorithms which relies on Markov chain Monte Carlo methods. Compared to existing personalised FL methods, the proposed methodology has important benefits: it is robust to client drift, practical for inference on new clients, and above all, enables uncertainty quantification under mild computational and memory overheads. We provide non-asymptotic convergence guarantees for the proposed algorithms and illustrate their performances on various personalised federated learning tasks.

## 1 Introduction

Federated learning (FL) is a recent machine learning paradigm in which distributed clients holding siloed data collaborate in solving a learning problem, usually under the coordination of a central server (Wang et al., 2021; Kairouz et al., 2021). One of the main focus of FL is on so-called *cross-device* applications where a large number of personal electronic devices such as mobile phones, wearable devices or home assistants collect and store data at the edges of a decentralised network (McMahan et al., 2017).

While standard FL methods (McMahan et al., 2017; Alistarh et al., 2017; Karimireddy et al., 2020; Horváth et al., 2019; Li et al., 2020) have focused on training a global model that can be applied to individual agents, the relevance of such inferences has recently been questioned due to statistical *heterogeneity* between clients. Indeed, the considered common model may not generalise well on a client with a local data distribution that differs significantly from the global data distribution, especially if that client has not participated in the training process. In fact, it might even be better for such clients to derive a local model from their own data set. To circumvent these issues, a number of

36th Conference on Neural Information Processing Systems (NeurIPS 2022).

*personalised federated learning* approaches have recently been proposed, that use local models to fit client-specific data distribution while capturing some shared knowledge via a federated scheme (Tan et al., 2022). Personalisation has previously been approached using multi-task learning (Smith et al., 2017), meta-learning (Jiang et al., 2019; Khodak et al., 2019), client clustering (Briggs et al., 2020), data interpolation (Mansour et al., 2020), model interpolation (Hanzely and Richtárik, 2020; Hanzely et al., 2020) or partially local models (Singhal et al., 2021; Collins et al., 2021). While these methodologies are promising, they only partially solve the personalisation problem in highly heterogeneous federated settings and have no means of quantifying uncertainty. In addition, cross-device FL also faces other important challenges such as (extreme) partial device participation, small local data sets, limited upload bandwidth and device capabilities (Kairouz et al., 2021). Addressing these problems for personalised FL requires new paradigms regarding how model knowledge is shared and personalisation is performed locally.

**Proposed Approach.** In this paper, we adopt a novel perspective to model the problem of personalised FL. This paradigm, called *mixed-effects modeling* (also known as multi-level or population approach) is widely used to analyse data that have a clustered or nested structure, as in medical or biological research where multiple observations per patient are available (Gelman and Hill, 2007; Long, 2011; Lavielle, 2014). Although the hierarchical structure of FL has already been noted (Plassier et al., 2021; Grant et al., 2018; Hong et al., 2022), the mixed-effects paradigm has interestingly never been considered. Leveraging this framework, we develop a new model for personalised FL called FedPop and propose an efficient computational solution to perform inference under this model. More precisely, we introduce a novel class of federated stochastic approximation algorithms based on parallel Markov Chain Monte Carlo (MCMC) methods. In the proposed framework, we also pay special attention to the cross-device setting by taking into account partial client participation, and by addressing the communication bottleneck with both multiple local updates and the use of lossy compression operators.

**Benefits.** Up to the authors' knowledge, FedPop is the first *personalised FL* approach that allows *cheap uncertainty quantification* with a theoretically-grounded methodology. The proposed framework also comes with other interesting properties. First, in contrast to most of personalised FL methods that only consider "fixed-effects" models (Collins et al., 2021; Hanzely et al., 2021; Smith et al., 2017), FedPop provides credibility information (via credibility regions) and allows more accurate inference for clients with small and heterogeneous local data via *partial pooling* (Gelman and Hill, 2007). In addition, inference for new clients which did not participate in the training phase can be easily performed by sampling from the prior over the local random effects. Second, contrary to existing Bayesian FL approaches that aim to provide credibility information by sampling from a target posterior distribution (Hong et al., 2022; Yoon et al., 2018; Vono et al., 2022; El Mekkaoui et al., 2021), FedPop allows to perform personalisation and cheaper on-device uncertainty quantification taking an empirical Bayes prediction approach. Finally, an important benefit of FedPop is its ability to allow for multiple local updates without suffering from the client-drift phenomenon (Karimireddy et al., 2020).

**Outline and Contributions.** Our contributions are fourfold. First, in Section 2, we propose a novel probabilistic methodology, which we call FedPop, to address personalisation under the cross-device FL paradigm. To perform efficient inference under this model, we introduce a general class of stochastic approximation algorithms based on MCMC. Second, we provide in Section 3 non-asymptotic convergence guarantees for the proposed methodology. Then, we perform in Section 4 a comparison between the proposed approach and exisiting works. Finally, we illustrate in Section 5 the benefits of our methodology on several federated learning benchmarks involving both synthetic and real data.

## 2 Proposed Approach

In this section, we present the statistical estimation problem we are considering and the proposed methodology called FedPop to address it.

**Problem Formulation.** We are interested in the *cross-device* FL setting involving a large number $b \in \mathbb{N}^*$ of clients, potentially unreliable *i.e.* not necessarily available at each communication round. These clients are assumed to own sensitive local data sets $\{D_i\}_{i \in [b]}$. In this framework, we aim to make both uncertainty quantification and personalised statistical inference by learning a local model

tailored to each client. To this end, and inspired by the population approach used in the biological and physical sciences (Lavielle, 2014), we consider mixed-effects modeling for each client leading to the local marginal likelihood function defined, for any $i \in [b]$, by

$$p(\mathrm{D}_i \mid \phi, \beta) = \int_{\mathbb{R}^d} p(\mathrm{D}_i \mid \phi, z^{(i)}) p(z^{(i)} \mid \beta) \mathrm{d}z^{(i)}, \tag{1}$$

where $\phi \in \Phi \subseteq \mathbb{R}^{d_\Phi}$ stands for a *fixed effect* and $\{z^{(i)}\}_{i \in [b]} \in \mathsf{Z}$, $\mathsf{Z} = \prod_{i=1}^b \mathbb{R}^d$, represent *random effects* aimed at explaining statistical heterogeneity between local data sets $\{\mathrm{D}_i\}_{i \in [b]}$.

The objective of the fixed (*i.e.* constant across all clients) part is to capture a common representation (*e.g.* same features across different classes of images) while the random part, which is typically low-dimensional, performs personalisation and is assumed to be drawn from a *population* prior whose variance aims at modeling data heterogeneity.

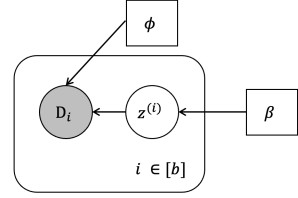

Figure 1 illustrates this statistical framework, referred to as FedPop, by showing its directed acyclic graph (DAG) where grey-filled shapes indicate observed variables, white-filled shapes unknown variables and squared shapes variables to be estimated.

Figure 1: DAG for FedPop.

When the size of the local data set $\mathrm{D}_i$ is small, this common prior leverages information from other clients to limit the risk of overfitting and is often called *partial pooling* in the multi-level statistical literature (Gelman and Hill, 2007, Section 12). Examples of model architectures involving $\phi$ and $\{z^{(i)}\}_{i \in [b]}$ include for instance *composition*-based architectures $p(\mathrm{D}_i \mid \phi, z^{(i)}) = p(\mathrm{D}_i \mid h_\phi \circ h_{z^{(i)}})$ where $h_\phi$ and $h_{z^{(i)}}$ are two neural networks (Collins et al., 2021; Arivazhagan et al., 2019). For the sake of generality, we propose to adopt a flexible energy-based prior distribution of the form for each $i \in [b]$,

$$p(z^{(i)} \mid \beta) = \frac{1}{Z(\beta)} \exp\left\{-E(z^{(i)}; \beta)\right\}, \text{ where } Z(\beta) = \int_{\mathbb{R}^d} \exp\left\{-E(z^{(i)}; \beta)\right\} \mathrm{d}z^{(i)}.$$

Here, $Z(\beta)$ is a normalising constant and $E(\cdot; \beta)$ represents an *energy* function, typically a neural network, parameterised by a set of parameters $\beta \in \mathsf{B} \subseteq \mathbb{R}^{d_\mathsf{B}}$ (LeCun et al., 2006). This framework is particularly interesting in the cross-device setting where the number of clients $b$ is large as it allows for efficient enrichment of the model. However, in the case where $b$ is small, the inference of the parameter $\beta$ is difficult. In this situation, a more pragmatic solution is to consider a common prior for the local random effects $\{z^{(i)}\}_{i \in [b]}$ which is held fixed, *i.e.* $p(z^{(i)} \mid \beta) \propto \exp\{-E(z^{(i)})\}$ for any $\beta \in \mathsf{B}$. Finally, for completeness, we allow the use of a prior model $p(\phi, \beta)$ for the hyperparameters $\{\phi, \beta\}$. Using Bayes' rule (Robert, 2001) and by denoting $\mathrm{D} = \sqcup_{i=1}^b \mathrm{D}_i$ the global data set, the posterior distribution associated with these hyperparameters admits a probability density function which can be written as

$$p(\phi, \beta \mid \mathrm{D}) \propto p(\phi, \beta) \prod_{i=1}^b \left[ \int_{\mathbb{R}^d} p(\mathrm{D}_i \mid \phi, z^{(i)}) p(z^{(i)} \mid \beta) \mathrm{d}z^{(i)} \right].$$

Set $\theta = \{\phi, \beta\} \in \Theta$ with $\Theta = \Phi \times \mathsf{B}$. In the sequel, we will be interested in solving the maximum a posteriori problem given by

$$\theta^\star \in \underset{\theta \in \Theta}{\arg\max} \ \log p(\phi, \beta \mid \mathrm{D}), \tag{2}$$

$$\log p(\phi, \beta \mid \mathrm{D}) = \log p(\phi, \beta) + \sum_{i=1}^b \left[ \log \int_{\mathbb{R}^d} p(\mathrm{D}_i \mid \phi, z^{(i)}) p(z^{(i)} \mid \beta) \mathrm{d}z^{(i)} \right] + C, \tag{3}$$

where $C \in \mathbb{R}$ is a constant independent of $\theta$. Once we have estimated $\theta^\star$, using an empirical Bayesian approach, we can perform "for free" on-device uncertainty quantification for each client $i \in [b]$ by sampling from the local posterior distribution $p(z^{(i)} \mid \mathrm{D}_i, \phi^\star, \beta^\star)$, which is typically designed to be low-dimensional.

**Algorithm.** To solve the optimisation problem (2), we can either use an *alternating maximisation* algorithm or perform global maximisation over $\Theta$. Since the former approach requires more upload

bandwidth, in this work we consider the second alternative which is more suitable for FL. The gradient of the objective function (3) being intractable, we propose to resort to the stochastic approximation framework (Robbins and Monro, 1951) which iteratively defines $(\phi_k, \beta_k)_{k \in \mathbb{N}}$, starting from any $(\phi_0, \beta_0) \in \Theta$, via the recursions for any $k \in \mathbb{N}$,

$$\beta_{k+1} = \Pi_{\mathsf{B}} \left( \beta_k + \eta_{k+1}^{(1)} \left[ \nabla_\beta \log p(\phi_k, \beta_k) + \sum_{i=1}^{b} g_k^{(i)}(\phi_k, \beta_k) \right] \right),$$

$$\phi_{k+1} = \Pi_{\Phi} \left( \phi_k + \eta_{k+1}^{(2)} \left[ \nabla_\phi \log p(\phi_k, \beta_k) + \sum_{i=1}^{b} h_k^{(i)}(\phi_k, \beta_k) \right] \right),$$

where $\Pi_{\mathsf{C}}$ denotes the projection onto $\mathsf{C} \in \{\Phi, \mathsf{B}\}$, $(\eta_k^{(1)}, \eta_k^{(2)})_{k \in \mathbb{N}^*}$ are sequences of step-sizes, and $\{g_k^{(i)} : i \in [b], k \in \mathbb{N}^*\}$ and $\{h_k^{(i)} : i \in [b], k \in \mathbb{N}^*\}$ are estimators of the intractable gradients $(\phi, \beta) \mapsto \nabla_\beta \log p(\mathrm{D}_i \mid \phi, \beta)$ and $(\phi, \beta) \mapsto \nabla_\phi \log p(\mathrm{D}_i \mid \phi, \beta)$ at $(\phi_k, \beta_k)$, where $p(\mathrm{D}_i \mid \phi, \beta)$ is defined in (1) for any $i \in [b]$.

The choices of the estimators $\{g_k^{(i)} : i \in [b], k \in \mathbb{N}^*\}$ and $\{h_k^{(i)} : i \in [b], k \in \mathbb{N}^*\}$ are motivated by the Fisher identity. More precisely, under mild regularity assumptions, and using the Lebesgue dominated convergence theorem, we have for any, $(\phi, \beta) \in \Theta, i \in [b]$

$$\nabla_\beta \log p(\mathrm{D}_i \mid \phi, \beta) = \int_{\mathbb{R}^d} \left[ \nabla_\beta \log p(\mathrm{D}_i, z^{(i)} \mid \phi, \beta) \right] p(z^{(i)} \mid \mathrm{D}_i, \phi, \beta) \mathrm{d}z^{(i)},$$

$$\nabla_\phi \log p(\mathrm{D}_i \mid \phi, \beta) = \int_{\mathbb{R}^d} \left[ \nabla_\phi \log p(\mathrm{D}_i, z^{(i)} \mid \phi, \beta) \right] p(z^{(i)} \mid \mathrm{D}_i, \phi, \beta) \mathrm{d}z^{(i)},$$

which suggests to consider

$$g_k^{(i)}(\phi, \beta) = \frac{1}{M} \sum_{m=1}^{M} \nabla_\beta \log p(Z_k^{(i,m)} \mid \beta), \tag{4}$$

$$h_k^{(i)}(\phi, \beta) = \frac{1}{M} \sum_{m=1}^{M} \nabla_\phi \log p(\mathrm{D}_i \mid Z_k^{(i,m)}, \phi), \tag{5}$$

where $M \in \mathbb{N}^*$ and $Z_k^{(i,1:M)} = (Z_k^{(i,m)})_{m \in [M]}$ are approximate samples from $p(z^{(i)} \mid \mathrm{D}_i, \phi, \beta)$. More precisely, we consider a family $\{Q_{\gamma,\theta}^{(i)} : \gamma \in (0, \bar{\gamma}], \theta \in \Theta\}$ where for any step-size $\gamma$, $Q_{\gamma,\theta}^{(i)}$ is a Markov kernel which targets a close approximation of $p(z^{(i)} \mid \mathrm{D}_i, \theta)$ with $\theta = \{\phi, \beta\}$. As an example, we can use overdamped Langevin dynamics (Roberts and Tweedie, 1996; Welling and Teh, 2011) to generate these samples. In this case, $Q_{\gamma,\theta}^{(i)}$ is associated with a Gaussian probability density function $q_{\gamma,\theta}^{(i)}(z^{(i)}, \cdot)$ with mean $z^{(i)} - \gamma \nabla_z \log p(z^{(i)} \mid \mathrm{D}_i, \theta)$ and variance $2\gamma \mathrm{I}_d$. Note that the number of Monte Carlo draws per iteration $k$ is considered constant here but we can easily generalise our scheme to the non-constant setting. In addition, our scheme can also be generalised by taking into account *stochastic* gradient estimators of (4) and (5). For the sake of simplicity, we present our approach with standard gradients.

In this framework, we present the main steps of the corresponding stochastic approximation algorithm, called `FedSOUK`, in Algorithm 1. Since we consider the *cross-device* federated setting, note that only a random subset $\mathsf{A}_{k+1}$ of active (*i.e.* available) clients communicates with the central server at each iteration $k \in \mathbb{N}$. In addition, due to limited upload bandwidth, the potentially high-dimensional gradient estimator (5) is compressed locally via an unbiased stochastic compression operator $\mathscr{C}_{k+1}$ before being sent to the central server (Alistarh et al., 2017; Philippenko and Dieuleveut, 2020).

**Stateful and Stateless Versions.** Depending on local memory constraints and the participation rate, we allow for a possible warm-start strategy across communication rounds to improve the convergence properties of the proposed algorithm so that the proposed algorithm becomes *stateful*, see steps $4 - 7$ in Algorithm 1. In cases the participation rate is very low (*e.g.* each client might only participates once to the training process), we replace this warm-start strategy by the initialisation $Z_k^{(i,0)} \sim p(\cdot \mid \beta_k)$ for any $i \in [b]$ and $k \in \{0, \ldots, K - 1\}$. This yields a *stateless* version of our algorithm more suitable to the cross-device setting. Obviously, compared to the previously proposed warm-start

strategy, the performances of Algorithm 1 will be affected negatively if we are using the same number of local iterations $M$. We end up with an interesting trade-off between local computations and communication: if client-server communication is a bottleneck, the stateless version of the algorithm allows to reduce the communication overhead at the price of longer sampling procedures on each client. Such analyses will be illustrated empirically in Section 5.

**Computation Complexity.** Compared to standard FL methods, our approach has an additional computational cost on the client side associated with Monte Carlo approximations $\{I_k^{(i)}\}$ and $\{J_k^{(i)}\}$. In practice, this cost is negligible. Indeed, in our experiments, we found that using a small value of $M \in [1, 10]$ was sufficient to obtain state-of-the-art results in terms of accuracy on the test dataset. We would like to emphasize that this additional computational cost has also two side advantages compared to existing FL approaches: (1) it allows us to communicate less frequently with the central server and (2) it allows us to converge faster when the number of local iterations increases since Monte Carlo approximation becomes better.

**Communication Overhead.** As pointed out in Table 1, our methodology `FedPop` improves upon existing FL approaches regarding the communication overhead. Indeed, `FedPop` offers the flexibility to use both compression for sending updates to the server, and multiple local steps to reduce the communication frequency. As such, depending on the bandwidth and local computational power, the practitioner can adapt the number of local iterations and the parameter of the compression operator. Up to our knowledge, this work is the first one combining compression and multiple local steps for personalised FL.

**Robustness to client drift.** For simplicity, we will take the example of `FedSOUL` (see Algorithm 1) which uses the Markov kernel associated with Langevin Monte Carlo to compute gradient estimates of the local marginal likelihood. However, our answer holds for general Markov kernels (adjusted or unadjusted). In this scenario, $M$ steps of Langevin Monte Carlo are performed on each device to draw samples $\{Z_k^{(i,m)}\}$ used to compute Monte Carlo estimates $\{I_k^{(i)}\}$ and $\{J_k^{(i)}\}$. Increasing the number of local steps $M$ does not slow down convergence but instead allows for more accurate Monte Carlo integration and hence better convergence properties. In contrast, the client drift phenomenon for classical FL approaches (e.g. `FedAvg` proposed in McMahan et al. (2017)) slows down convergence as the number of local iterations $M$ increases.

**Simple inference on new clients.** Typical personalized FL approaches such as `DITTO` or `FedRep` require additional local training for inference on new clients. In contrast, the proposed methodology `FedPop` allows for a cheaper two-step approach once we have estimated $\theta^\star = (\phi^\star, \beta^\star)$, as detailed below for a new client $b + 1$ with feature vector $x$:

1. Sample $\{Z_{b+1}^{(l)}\}_{l \in [L]}$ in a i.i.d. manner.

2. Estimate the posterior predictive function by $p(\cdot \mid x) \approx L^{-1} \sum_{l=1}^{L} p(\cdot \mid \phi^\star, Z_{b+1}^{(l)}, x)$.

The prior $p(z_{b+1} \mid \beta^\star)$ is typically chosen so that sampling is computationally cheap, *e.g.* a Gaussian with diagonal covariance matrix as in our experiments, see Section 5.

## 3   Theoretical Guarantees

In this section, we present non-asymptotic convergence guarantees for Algorithm 1 when the family of Markov kernels $\{Q_{\gamma,\theta}^{(i)} : \gamma \in (0, \bar{\gamma}], \theta \in \Theta, i \in [b]\}$ is associated to unadjusted, *i.e.* without Metropolis acceptance step, overdamped Langevin dynamics (Durmus and Moulines, 2017; Dalalyan, 2017). The bounds we derive allow to showcase explicitly the impact of FL constraints, namely partial participation and compression. Results for general unadjusted Markov kernels are postponed to the supplement.

To show our theoretical results and resort to standard assumptions made in the stochastic approximation literature, we consider a minimisation problem and rewrite the opposite of the objective function (3) for any $\theta \in \Theta$ as

$$f(\theta) = b^{-1} \sum_{i=1}^{b} f_i(\theta), \quad \text{where } f_i(\theta) = -\log p(\phi, \beta) - b \log p(D_i \mid \phi, \beta). \qquad (6)$$

---

**Algorithm 1** FL via Stochastic Optimisation using Unadjusted Kernel (`FedSOUK`)

---

1: **Input:** nb. outer iterations $K$, nb. local iterations $M$, Markov kernels $\{Q^{(i)}_{\gamma,\theta}\}_{\gamma,\theta,i}$, step-sizes $\{\eta^{(1)}_k, \eta^{(2)}_k\}_{k\in[K], i\in[b]}$ and initial points $Z^{(0)}_0 \in \mathbb{R}^d$, $\beta_0 \in \mathsf{B}$ and $\phi_0 \in \Phi$.
2: **for** $k = 0$ to $K-1$ **do**
3:   Server sends $\{\beta_{k+1}, \phi_{k+1}\}$ to clients belonging to $\mathsf{A}_{k+1}$.
4:   **for** $i \in \mathsf{A}_{k+1}$ // On active clients $\mathsf{A}_{k+1}$ **do**
5:    // Warm-start of the SA scheme if possible
6:    **if** $k \geq 1$ **then**
7:     Set $Z^{(i,0)}_k = Z^{(i,M)}_{k-1}$.
8:    **end if**
9:    // Computation of key quantities using MCMC
10:    **for** $m = 0$ to $M-1$ **do**
11:     Draw $Z^{(i,m+1)}_k \sim Q^{(i)}_{\gamma,\theta_k}\left(Z^{(i,m)}_k, \cdot\right)$.
12:     // For Langevin dynamics
13:     // Draw $\xi^{(i,m+1)}_k \sim \mathrm{N}(0_d, \mathrm{I}_d)$.
14:     // Set $Z^{(i,m+1)}_k = Z^{(i,m)}_k + \gamma \nabla_z \log p(Z^{(i,m)}_k \mid \mathrm{D}_i, \phi_k, \beta_k) + \sqrt{2\gamma}\xi^{(i,m+1)}_k$.
15:    **end for**
16:    // Communication with the server
17:    Set $I^{(i)}_k = \frac{1}{M}\sum_{m=1}^{M} \nabla_\beta \log p\left(Z^{(i,m)}_k \mid \beta_k\right)$.
18:    Set $J^{(i)}_k = \frac{1}{M}\sum_{m=1}^{M} \nabla_\phi \log p\left(\mathrm{D}_i \mid Z^{(i,m)}_k, \phi_k\right)$.
19:    Send $I^{(i)}_k$ and $\mathscr{C}_{k+1}\left(J^{(i)}_k\right)$ to the central server.
20:   **end for**
21:   Set $\beta_{k+1} = \Pi_{\mathsf{B}}\left(\beta_k + \eta^{(1)}_{k+1}\left[\nabla_\beta \log p(\phi_k, \beta_k) + \frac{b}{|\mathsf{A}_{k+1}|}\sum_{i\in\mathsf{A}_{k+1}} I^{(i)}_k\right]\right)$.
22:   Set $\phi_{k+1} = \Pi_{\Phi}\left(\phi_k + \eta^{(2)}_{k+1}\left[\nabla_\phi \log p(\phi_k, \beta_k) + \frac{b}{|\mathsf{A}_{k+1}|}\sum_{i\in\mathsf{A}_{k+1}} \mathscr{C}_{k+1}\left(J^{(i)}_k\right)\right]\right)$.
23: **end for**
24: **Output:** $\{\phi_K, \beta_K\}$ and samples $\{Z^{(1:b,m)}_{K-1}\}^M_{m=1}$.

---

**Non-Asymptotic Convergence Bounds.** For the sake of better readability, we only detail in the main paper assumptions regarding the objective function, compression operators and the partial participation scenario. Technical assumptions related to the Markov kernels $\{Q^{(i)}_{\gamma,\theta}\}$ are postponed to the supplement. In spirit, we require, for any $i \in [b], \theta \in \Theta$ and $\gamma$, that $Q^{(i)}_{\gamma,\theta}$ satisfies some ergodic condition and can provide samples sufficiently close to the local posterior distribution $p(z^{(i)} \mid \mathrm{D}_i, \theta)$. For the sake of simplicity, we also assume that for any $k \in \mathbb{N}^*, \eta^{(1)}_k = \eta^{(2)}_k = \eta_k$, see Algorithm 1.

We make the following assumptions on $\Theta$ and the family of functions $\{f_i : i \in [b]\}$.

**H1.** $\Theta$ *is convex, closed subset of* $\mathbb{R}^{d_\Theta}$ *and* $\Theta \subset \mathrm{B}(0, R_\Theta)$ *for* $R_\Theta > 0$.

**H2.** *For any* $i \in [b]$, *the following conditions hold.*

*(i) The function* $f_i$ *defined in* (6) *is convex.*
*(ii) There exist an open set* $\mathsf{U} \in \mathbb{R}^{d_\Theta}$ *and* $L_f > 0$ *such that* $\Theta \subset \mathsf{U}$, $f_i \in \mathrm{C}^1(\mathsf{U}, \mathbb{R})$ *and for any* $\theta_1, \theta_2 \in \Theta$,

$$\|\nabla f_i(\theta_2) - \nabla f_i(\theta_1)\| \leq L_f \|\theta_2 - \theta_1\| .$$

The assumption below requires compression operators $\{\mathscr{C}_k\}_{k\in\mathbb{N}^*}$ to be unbiased and to have a bounded variance. Such an assumption is for instance verified by stochastic quantisation operators, see Alistarh et al. (2017).

**H3.** *The compression operators* $\{\mathscr{C}_k\}_{k\in\mathbb{N}^*}$ *are independent and satisfy the following conditions.*

*(i) For any* $k \in \mathbb{N}^*$, $v \in \mathbb{R}^d$, $\mathbb{E}[\mathscr{C}_k(v)] = v$.
*(ii) There exists* $\omega \geq 1$, *such that for any* $k \in \mathbb{N}^*$, $v \in \mathbb{R}^d$, $\mathbb{E}[\|\mathscr{C}_k(v) - v\|^2] \leq \omega \|v\|^2$.

We finally assume that each client has probability $p \in (0, 1]$ to be active at each communication round. We would like to point out that this partial participation assumption can be associated to a specific compression operator satisfying **H**3.

**H4.** *For any $k \in \mathbb{N}^*$, $\mathsf{A}_k = \{i \in [b] : B_{i,k} = 1\}$ where for any $i \in [b]$, $\{B_{i,k} : k \in \mathbb{N}^*\}$ is a family of i.i.d. Bernouilli random variables with success probability $p \in (0, 1]$.*

Under these assumptions, the next result establishes that $(\bar{\theta}_k)_{k \in \mathbb{N}}$ defined by $\bar{\theta}_k = \sum_{j=1}^{k} \eta_j \theta_j / (\sum_{j=1}^{k} \eta_j)$ converges towards an element of $\arg\min_{\Theta} f$.

**Theorem 1.** *Assume **H**1-**H**4 along with **A**8 detailed in the supplement and let for any $k \in [K]$, $\eta_k \in (0, 1/L_f]$. Then, for any $K \in \mathbb{N}^*$, we have*

$$\mathbb{E}\left[f(\bar{\theta}_k) - f(\theta_\star)\right] \leq \mathbb{E}\left[\frac{\sum_{k=1}^{K} \eta_k \{f(\theta_k) - f(\theta_\star)\}}{\sum_{k=1}^{K} \eta_k}\right] \leq A(\gamma) + \frac{E_K}{\sum_{k=1}^{K} \eta_k},$$

*where $E_K$ depends linearly on $(\omega/p) \sum_{k=1}^{K} \eta_k^2$; and $A(\gamma) = C\gamma^\alpha$ with $\alpha > 0$ and $C$ is independent of $\omega, p$ and $(\eta_k)$. Closed-form formulas for these constants are provided in the supplement.*

An interesting feature of Algorithm 1 is that convergence towards a minimum of $f$ is possible and the impact of partial participation and compression vanishes when $\lim_{k \to \infty} \eta_k = 0$. More precisely, $\limsup_{k \to \infty} E_K / (\sum_{k=1}^{K} \eta_k) = 0$ and $\lim_{\gamma \to 0^+} A(\gamma) = 0$ which shows that we can tend towards a minimum of $f$ with arbitrary precision $\epsilon > 0$ by setting the step-size $\gamma$ to a small enough value.

## 4 Related Works

As pointed out in Section 1, many different approaches have been proposed to address personalisation and uncertainty quantification under the federated learning paradigm. This section reviews the main related existing lines of research and shows that the proposed methodology provides many benefits; see Table 1. Interestingly, we also show that `FedPop` encompasses some of the existing FL models.

**Bayesian FL.** One of our main motivations is the possibility to perform grounded uncertainty quantification in FL by resorting to the Bayesian paradigm. In the recent years, many works have suggested to adapt serial workhorses stochastic simulation approaches such as MCMC or variational inference to the FL setting (Chen and Chao, 2020; Liu and Simeone, 2021b,a; Vono et al., 2022; El Mekkaoui et al., 2021; Corinzia et al., 2019; Bui et al., 2018; Plassier et al., 2021; Deng et al., 2021a). Although some of these approaches address important FL challenges such as the communication bottleneck, partial participation or limited computational device resources, they are not suitable for uncertainty quantification in the cross-device FL scenario. Indeed, all these approaches assume that the posterior distribution targeted by each client is parametrised by a single potentially high-dimensional parameter of size $d_\Phi + d$, see (1). This prevents a sufficient number of samples from being stored locally to perform uncertainty quantification and Bayesian model averaging, especially when the model is a large neural network. In contrast, our approach decouples this unique high-dimensional parameter into a fixed part $\phi$ and a low-dimensional random part $z^{(i)}$, significantly reducing the memory footprint of local sample storage.

In addition, Bayesian FL methods aim at sampling a random parameter from a target probability distribution where $\pi(\theta) \propto e^{-f(\theta)}$ where $f = (1/b) \sum_{i=1}^{b} f_i$ denotes the negative log-likelihood associated to the $i$-th client. On the other hand the proposed methodology considers a mixed-effects modeling approach where parameters are divided into two categories: a fixed component and a random one for each client. As such, the mixed-effects approach is in essence an empirical Bayesian/marginal likelihood approach (Casella, 1985; Robbins, 1992). It corresponds to a hierarchical model that aims to combine the modeling flexibility and uncertainty assessment of Bayesian inference with computational pragmatism. More precisely, a part of the parameters (fixed-effects ) are estimated via marginal likelihood maximisation and the rest (random effects) using common Bayesian techniques, which are in most cases low dimensional. As a result, up to our knowledge, the model and approach that we propose is novel in FL and comes with many benefits as shown in Table 1.

**Personalised FL.** Beside uncertainty quantification, we also aim at providing each client with a dedicated personalised model. Among the numerous existing personalised FL approaches, those related to `FedPop` can be broadly classified into two groups: *meta-learning* and *partially local*

Table 1: Overview of the main existing personalised FL (top rows) and Bayesian FL (bottom rows) approaches related to the proposed framework. Column "PP" refers to partial participation, "perso." to personalised approaches, "bounds" to available convergence guarantees, "UQ" to available uncertainty quantification, "com." to the scheme (multiple local steps and/or compression) used to address the communication bottleneck and "memory" to the client memory footprint where $M$ stands for the number of samples.

| METHOD | PP | PERSO. | BOUNDS | UQ | COM. | MEMORY | FEDPOP INSTANCE |
|--------|----|--------|--------|----|------|--------|-----------------|
| PER-FEDAVG | ✓ | ✓ | ✓ | ✗ | LOCAL STEPS | $d + d_\Phi$ | ✗ |
| pFEDME | ✗ | ✓ | ✓ | ✗ | LOCAL STEPS | $d + d_\Phi$ | ✗ |
| FEDREP | ✓ | ✓ | ✓ | ✗ | LOCAL STEPS | $d + d_\Phi$ | ✓ |
| DITTO | ✓ | ✓ | ✓ | ✗ | LOCAL STEPS | $d + d_\Phi$ | ✗ |
| LG-FEDAVG | ✓ | ✓ | ✓ | ✗ | LOCAL STEPS | $d + d_\Phi$ | ✗ |
| QLSD | ✓ | ✗ | ✓ | ✓ | COMPRESSION | $M(d + d_\Phi)$ | ✗ |
| FSGLD | ✗ | ✗ | ✓ | ✓ | LOCAL STEPS | $M(d + d_\Phi)$ | ✗ |
| FEDBE | ✓ | ✗ | ✗ | ✓ | LOCAL STEPS | $M(d + d_\Phi)$ | ✗ |
| DG-LMC | ✗ | ✗ | ✓ | ✓ | LOCAL STEPS | $M(d + d_\Phi)$ | ✓ |
| FEDPOP | ✓ | ✓ | ✓ | ✓ | BOTH | $Md + d_\Phi$ | – |

*methods.* Meta-learning based FL methods aim at training a global model conducive to fast training of personalised models. Such a goal can be achieved, for example, by local fine-tuning (Fallah et al., 2020), regularisation of local models towards their average (Hanzely and Richtárik, 2020; Hanzely et al., 2021) – or the opposite (Li et al., 2021), and model interpolation (Liang et al., 2019). On the other hand, FL methods based on partial decoupling take an approach similar to ours by splitting the initial model into a backbone component and a local one aimed at personalisation (Collins et al., 2021; Arivazhagan et al., 2019; Pillutla et al., 2022). This partial decoupling could also enhance privacy as discussed in Singhal et al. (2021). The main difference with FedPop is that such approaches based on empirical risk minimisation cannot provide credibility information.

**FedPop: A Compromise between Standard and Personalised FL.** Interestingly, we show here that the FedPop framework allows existing FL approaches to be retrieved in certain regimes. To this end, we assume that the prior $p(z^{(i)} \mid \beta)$ is Gaussian with mean $\mu$ and covariance matrix $\sigma^2 I_d$ so that $\beta = \{\mu, \sigma\}$. If $\sigma \to 0^+$, then this Gaussian prior tends towards the Dirac distribution centered at $\mu$ and the local likelihood becomes $p(D_i \mid \phi, \mu)$, which corresponds to the local objective of standard FL approaches such as FedAvg (McMahan et al., 2017). On the other hand, when $\sigma \to \infty$, no common information $\beta$ is used to locally regress $z^{(i)}$ and we end up with the FedRep algorithm (Collins et al., 2021). This shows that FedPop stands for a subtle compromise between standard and personalised FL which should benefit clients with small data sets by pooling information via a common prior. Finally, in the extreme scenario where $\phi$ is the null vector, our approach amounts to the Bayesian FL approach DG-LMC proposed in Plassier et al. (2021).

# 5 Numerical Experiments

In this section, we illustrate the benefits of our methodology on several FL benchmarks associated to both synthetic and real data. Since existing Bayesian FL approaches are not suited for personalisation (see Table 1), we only compare the performances of Algorithm 1 with personalised FL methods. In all our experiments, we use overdamped Langevin dynamics to sample locally and call this specific instance of Algorithm 1, FedSOUL. In addition, we set $p(z^{(i)} \mid \beta) = N(\mu, \sigma^2 I_d)$ with $\beta = \{\mu, \sigma\}$ for simplicity. To be comparable with existing personalised FL approaches that only consider periodic communication via multiple local steps, we do not resort to the proposed compression mechanism although the latter could be of interest for real-world applications. Additional experiments and details about experimental design are provided in the supplement.

**Synthetic Data.** We start by showcasing the benefits of FedSOUL for clients having small and highly heterogeneous data sets as pointed out in Section 1 and Section 2. To this end, we consider a similar experimental setting as in Collins et al. (2021) where synthetic observations $\{y_j^{(i)}\}_{j \in [N_i]} \in D_i$

are generated via the following procedure: $x_j^{(i)} \sim \mathrm{N}(0_k, \mathrm{I}_k)$ and $y_j^{(i)} \sim \mathrm{N}(z_{\mathrm{true}}^{(i)} \phi_{\mathrm{true}}^\top x_j^{(i)}, 0.1)$. The ground-truth parameters $z_{\mathrm{true}}^{(i)} \in \mathbb{R}^d$ and $\phi_{\mathrm{true}} \in \mathbb{R}^{k \times d}$ have been randomly generated beforehand with $(d, k) = (2, 20)$. Compared to Collins et al. (2021), we use heterogeneous data partitions across clients so that 90% of the $b = 100$ clients have small data sets of size 5 and the remaining 10% have data sets of size 10. We compare our results with FedRep (Collins et al., 2021) and FedAvg (McMahan et al., 2017) since they stand for two limiting instances of the proposed methodology, see Section 4 and Gelman and Hill (2007, Section 12).

Figure 2 compares the different approaches by computing the principle angle distance[1] (respectively the $\ell_2$ norm) between $\phi_{\mathrm{true}}$ (respectively $z_{\mathrm{true}}^{(i)}$) and its estimated value; the lesser the better. In contrast to its main competitors and based on both metrics, FedSOUL provides an impressive improvement. This illustrates the benefits of the introduction of a common prior $p(z^{(i)} \mid \beta)$ which allows to pre-

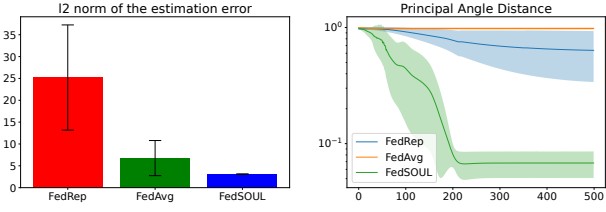

Figure 2: Small data sets - synthetic data.

vent from overfitting on clients with small data sets while performing personalisation. Additional results with other choices for $(b, d, k)$ and data partitioning strategies are available in the supplement.

Moreover, to compare our algorithm with a non-FL setting, we perform a non-distributed and non-federated stochastic approximation algorithm to find $\theta^*$ using a large number of iterations to get an accurate approximation of the optimal parameter $\theta^*$. Then, we use FedPop to obtain an estimate $\tilde{\theta}^*$ and measure the relative error in $l_2$- distance between $\theta^*$ and $\tilde{\theta}^*$. For some outer iterations $T = 100$, the relative error was less than $10^{-3}$, which illustrates the relevance of our theoretical results. We also test the performances of the proposed approach when the warm-start strategy is not used. In this case, we have to set $M = 50$ to achieve the same performances as in the stateful variant of FedSOUL.

**Real Data.** We consider now real image data sets, namely CIFAR-10 and CIFAR-100 (Krizhevsky, 2009). For our likelihood model defined by $p(\mathrm{D}_i \mid \phi, z^{(i)})$, we use 5-layer convolutional neural networks and perform personalisation for the last layer. We set $b = 100$ for convenience and control data heterogeneity by assigning to each client $N_i$ images belonging to only $S$ different classes.

*Small data sets.* Under this setting, we first consider (10%, 50%, 90%) of clients having small data sets of size either $N_i = 5$ or $N_i = 10$; while remaining clients have larger data sets of size $N_i = 25$. We compare our approach with FedRep since it stands for the state-of-the-art personalised FL approach. The algorithms are trained fulfilling the same computational budget. Figure 3 shows the average accuracy across clients for the two approaches on both CIFAR-10 and CIFAR-100.

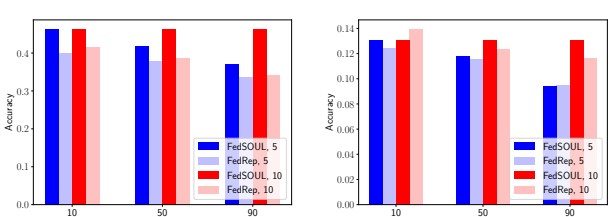

Figure 3: (right) CIFAR-10 with $S = 5$ and (left) CIFAR-100 with $S = 20$. The $x$-axis refers to the percentage of clients having $N_i \in \{5, 10\}$ images.

We can see that FedSOUL is consistently better than FedRep over different configurations.

*Full data sets.* In addition to show that the proposed approach achieves state-of-the-art performances on small data sets (which is common in the cross-device scenario), we now illustrate that FedSOUL is also competitive on larger data sets. To this end, we use all training images in CIFAR-10 and CIFAR-100 image data sets and consider the same data partitioning as in Collins et al. (2021). More precisely, in this case the number of observations and the number of classes per client are uniformly shared over the clients. Table 2 shows our results in comparison with state-of-the-art personalised FL approaches. We can see that that our model outperforms other methods on both CIFAR-10

---

[1] defined in (Collins et al., 2021, Definition 1)

Table 2: Real data - Full data sets. Accuracy (in %) on test samples. `FedAvg` and `SCAFFOLD` are not personalised FL approaches but stand for well-known FL benchmarks.

| | CIFAR-10 | | CIFAR-100 | |
|---|---|---|---|---|
| (# clients $b$, # classes per client $S$) | (100, 2) | (100, 5) | (100, 5) | (100, 20) |
| Local learning only | 89.79 | 70.68 | 75.29 | 41.29 |
| `FedAvg` (McMahan et al., 2017) | 42.65 | 51.78 | 23.94 | 31.97 |
| `SCAFFOLD` (Karimireddy et al., 2020) | 37.72 | 47.33 | 20.32 | 22.52 |
| `LG-FedAvg` (Liang et al., 2019) | 84.14 | 63.02 | 72.44 | 38.76 |
| `Per-FedAvg` (Fallah et al., 2020) | 82.27 | 67.20 | 72.05 | 52.49 |
| `L2GD` (Hanzely and Richtárik, 2020) | 81.04 | 59.98 | 72.13 | 42.84 |
| `APFL` (Deng et al., 2021b) | 83.77 | 72.29 | 78.20 | 55.44 |
| `DITTO` (Li et al., 2021) | 85.39 | 70.34 | 78.91 | 56.34 |
| `FedRep` (Collins et al., 2021) | 87.70 | 75.68 | 79.15 | 56.10 |
| `FedAvg + fine-tuning (FT)` | 85.63 | 71.32 | 79.03 | 56.19 |
| FedSOUL (this paper) | 91.12 | 79.48 | 79.56 | 59.73 |

and CIFAR-100 by a large margin. Additional results with other personalised FL algorithms are postponed to the supplement.

**Uncertainty Quantification on Real Data.** As highlighted in Table 1, one advantage of the proposed approach compared to existing personalised FL methods is the ability to perform uncertainty quantification by sampling locally from the posterior $p(z^{(i)} \mid \mathrm{D}_i, \phi_K, \beta_K)$, see Algorithm 1. We illustrate this feature by computing on CIFAR-10 calibration curves and scores (*e.g.* expected calibration error aka ECE) on a specific client; and by performing an out-of-distribution analysis on MNIST/FashionMNIST data sets. Figure 4 shows that the proposed approach provides relevant uncertainty diagnosis. Additional results on uncertainty quantification can be found in the supplement.

## 6   Conclusion

In this paper, we proposed a general Bayesian methodology based on a natural mixed-effects modeling approach to model personalisation in federated learning. Our FL method is the first that allows for both personalisation and cheap uncertainty quantification for (cross-device) federated learning. By introducing a common prior on the local parameters, we tackle the local overfitting problem in the scenario where clients have highly heterogeneous and small data sets. In addition, we have shown that the proposed approach has favorable convergence properties. Some limitations of `FedPop` pave the way for more advanced personalised FL approaches. As an ex-

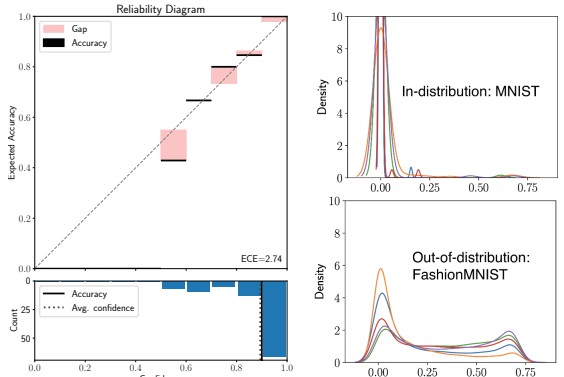

Figure 4: (right) Calibration on CIFAR-10 for a specific client and (left) OOD analysis with MNIST training & FashionMNIST inference – one curve corresponds to one client.

ample, our model does not allow for training heterogeneous architectures across clients because of the introduced common prior, and only satisfy first-order privacy guarantees. Regarding the latter, further works include for instance deriving differentially private versions of our framework.

## Acknowledgments and Disclosure of Funding

The authors acknowledge the Lagrange Mathematics and Computing Research Center for supporting the project. The development of the algorithm and conducting experiments (Section 5) was supported by Russian Science Foundation grant 20-71-10135.

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
