# OpenReview forum: "FedPop: A Bayesian Approach for Personalised Federated Learning"
_NeurIPS.cc/2022/Conference — NeurIPS 2022 Accept_

### Official Review · Reviewer_8AYd · 2022-07-11

**Rating:** 6
**Confidence:** 3
**Soundness:** 2 fair
**Presentation:** 3 good
**Contribution:** 3 good

**Summary:**

This paper proposes a new algorithm FedPop for personalized federated learning by modeling it as a mixed-effects probabilistic problem. Specifically, a client’s local distribution depends on a fixed-effect part (constant for all clients) and a random-effects part (captures the heterogeneity).  FedPop is developed by maximizing a posteriori probability via stochastic approximation. The algorithm works as follows: in each round, the participating clients will use MCMC to generate samples of the random-effects random variable, use them to compute approximate gradients, and send the updates to the server. A convergence bound is established for the proposed algorithm under appropriate assumptions. The authors empirically show that the proposed algorithm outperforms related works over a synthetic dataset, and two versions of CIFAR datasets (CIFAR-10 and CIFAR-100).

**Questions:**

See my detailed comments above for questions and suggestions, especially my concerns on application in the large-scale cross-device setting and the missing baseline FedAvg+FT with only fine-tuning the last layer (note: both the fine-tuning learning rate and number of epochs are hyperparameters that should be properly tuned from data).

**Limitations:**

The authors pointed out the limitations and future work in Section 6.

**Strengths And Weaknesses:**

Strength:

- The paper is well-written and easy to read.

- Modeling the personalized federated learning problem using the mix-effects paradigm seems new in the literature. The proposed algorithm allows each client to learn a personalized model and at the same time perform uncertainty quantification.

- In addition to the empirical comparisons, the authors provide theoretical guarantees and establish convergence bounds for the proposed algorithm.

Weakness:

- My main concern is that while the paper claims that the proposed algorithm is designed for the cross-device FL setting, it is unclear to me that FedSOUK (Algorithm 1) is indeed suitable for real-world cross-device applications. Cross-device settings typically involve a very large population (e.g., hundreds of millions mobile devices), and it is common that a device only participates at most once during the training process [1]. As a result, cross-device setting usually needs *stateless* algorithms rather than *stateful* algorithms. The main difference is whether the algorithm relies on clients maintaining local states from one round to another (see section 5.1 of [2] for why the stateful algorithm performs poorly in the cross-device setting). The proposed algorithm FedSOUK (Algorithm 1) is a stateful algorithm, because the participating clients warm-start MCMC from the $Z$ value from the previous round. In the real-world cross-device setting where most clients participate only once in training, this means that warm-starting is impossible, will this constraint affect the performance of the proposed algorithm? In that case, will each client need to draw a lot of $Z$ samples locally (how do we set $M$ in practice)? All the experiments shown in the paper are using datasets with a small number of clients (b=100), which is a scale of typical *cross-silo* federated settings [1]. It is thus not convincing to me that the proposed algorithm can perform similarly well in the real-world cross-device setting with millions of model devices and really low sampling rate per round (where most clients participate at most once in the entire training process).

- Other comments/questions:
  - FedAvg + fine-tuning (FT) a simple, effective, and stateless personalization algorithm (see, e.g., [3][4]) used in real-world cross-device FL applications (e.g., [5]). It is better to include that in Table 2 for a more comprehensive comparison (similar to Table 1 in [6]). On image datasets like CIFAR, fine-tuning only the last layer is usually better than fine-tuning the entire model. Besides, use the last layer as the personalization layer is also consistent with the proposed algorithm FedSOUL. Therefore, it is more convincing to add the baseline experimental results of FedAvg+FT with only fine-tuning the last layer to Table 2 (note: both the fine-tuning learning rate and number of epochs are hyperparameters that should be tuned from data).

  - The posterior distribution $p(\phi, \beta | D)$ equation (Line 117) does not contain $p(D)$ on the right-hand side, why is that?

  - In Algorithm 1, the step of server “Send {\beta_{k+1}, \phi_{k+1}} to clients belonging to A_{k+1}” happens at the end of every round, why does server not broadcasting the values to the participating clients at the beginning of every round?

[1] Kairouz et al., "Advances and open problems in federated learning." Foundations and Trends in Machine Learning,  (2021): 1-210.

[2] Reddi et al., “Adaptive federated optimization.”, ICLR, 2021.

[3] Wu et al., “Motley: Benchmarking Heterogeneity and Personalization in Federated Learning”, arXiv: 2206.09262, 2022.

[4] Matsuda et al., “An Empirical Study of Personalized Federated Learning”, arXiv:2206.13190, 2022.

[5] Wang et al., "Federated evaluation of on-device personalization." arXiv preprint arXiv:1910.10252, 2019.

[6] Collins et al. "Exploiting shared representations for personalized federated learning." ICML, 2021.

-------------------------update after the author rebuttal-------------------------------
The reviewers showed additional experimental (and promising) results in the cross-device setting: https://github.com/anonneurips1/rebuttal_neurips/tree/main
One limitation of the proposed method is that it requires each client to maintain $M$ samples each with dim($z$), where $M$ is around 50 in the cross-device setting. If dim($z$) is similar to dim($\phi$) (which is the case for the StackOverflow experiments), this means that the memory cost is 50 times that of other personalized algorithms such as FedAvg with fine-tuning.

I decided to increase my score from 5 to 6 because: 1) the memory&computation limitation is common for other Bayesian FL algorithms; 2) the proposed algorithm seem a strong baseline when the model size is small (or when the client's local resources are not constrained, such as in the cross-silo setting).

---

> ### Author Response · Authors · 2022-08-02
> **Response to Reviewer 8AYd**
>
> We would like to thank the anonymous reviewer for the constructive feedback and relevant remarks, notably on the cross-device scenario.
> Please find below our answer to all your questions and concerns.
>
> **1. My main concern is that while the paper claims that the proposed algorithm is designed for the cross-device FL setting, it is unclear to me that FedSOUK (Algorithm 1) is indeed suitable for real-world cross-device applications.**
>
> Thanks for this interesting and relevant remark. We will answer it point by point in what follows.
>
> - **The proposed algorithm FedSOUK (Algorithm 1) is a stateful algorithm, because the participating clients warm-start MCMC from the Z value from the previous round.**
>
>    We agree with the reviewer that stateless FL methods are indeed required in the cross-device scenario when the participation rate is very low.
>     When introducing the warm-start strategy to speed-up convergence, we explicitly pointed out that this strategy is optional (see l.153 and l.154 and in Algorithm 1). Nevertheless, we completely agree with the reviewer that we should have emphasized this point and detail the consequences of maintaining a local state on each device for cross-device applications. This will be done in the revised version of the manuscript.
>
> - **In the real-world cross-device setting where most clients participate only once in training, this means that warm-starting is impossible, will this constraint affect the performance of the proposed algorithm?
>     In that case, will each client need to draw a lot of Z samples locally (how do we set M in practice)?**
>
>     As you are pointing out, the warm-start strategy cannot be used in this cross-device scenario.
>     As a result, we can propose a *stateless* way to initialize the Markov chain on each device corresponding to sampling from the prior distribution i.e. $Z_k^{(i,0)} \sim p(\cdot \mid \beta_k)$.
>     Obviously, compared to the previously proposed warm-start strategy, the performances of Algorithm 1 will be affected negatively if we are using the same number of local iterations $M$.
>     We ran several experiments to check if we are capable to reproduce our previous results, using a stateless (without warm-starts) version of our algorithm.
>     As expected, obtaining results similar to the warm-start scenario requires a larger number of inner iterations $M$ because there is now a transient phase at each iterations of the algorithm. Concretely, our new experiments shows that $M$ has to be set to 50 in this case (in contrast to 10 in the warm-start scenario).
>     We end up with an interesting trade-off between local computations and communication: If client-server communication is a bottleneck, the stateless version of the algorithm allows to reduce the communication overhead at the price of longer sampling procedures on each client.
>
> - **All the experiments shown in the paper are using datasets with a small number of clients (b=100), which is a scale of typical cross-silo federated settings [1]. It is thus not convincing to me that the proposed algorithm can perform similarly well in the real-world cross-device setting.**
>
>     We agree with the reviewer. Following your other comments, we performed additional experiments in the cross-device setting by using the EMNIST and StackOverflow experiments described in [3]. Even in this setting, our method yields state-of-the-art results compared to other competing FL approaches. These results can be found following the link: https://github.com/anonneurips1/rebuttal_neurips/tree/main.
>
>     [3] Wu et al., “Motley: Benchmarking Heterogeneity and Personalization in Federated Learning”, arXiv: 2206.09262, 2022.
>
> **2. FedAvg + fine-tuning (FT) a simple, effective, and stateless personalization algorithm (see, e.g., [3][4]) used in real-world cross-device FL applications (e.g., [5]). It is better to include that in Table 2 for a more comprehensive comparison (similar to Table 1 in [6]).**
>
> Thanks for this remark. We added this baseline to our Table, see https://github.com/anonneurips1/rebuttal_neurips/tree/main.
> Even by taking into account this baseline, our approach still stands as a competing approach while allowing uncertainty quantification via Bayesian inference.
>
> **3. The posterior distribution equation (Line 117) does not contain $p(D)$ on the right-hand side, why is that?**
>
> This is indeed a typo and we should have used the $\propto$ sign instead of an equality. This will be fixed in the revised version of the manuscript.
>
> **4. In Algorithm 1, the step of server “Send {$\beta_{k+1}, \phi_{k+1}$} to clients belonging to $A_{k+1}$” happens at the end of every round, why does server not broadcasting the values to the participating clients at the beginning of every round?**
>
> This is an arbitrary choice but we agree with the reviewer that moving this step to the beginning of each communication round is more coherent. This will be fixed in the revised version of the manuscript.

---

> > ### Comment · Reviewer_8AYd · 2022-08-08
> > **Thanks for the extra cross-device experiments**
> >
> > Thanks for the additional experiments in the cross-device setting. The results look promising. Can you give more information about the parameters used in those extra experiments (on EMNIST and StackOverflow)? In particular, what are $\phi$, $z$, $\beta$, their dimensions, and the value of $M$? As pointed out by the author, a large value of $M$ is needed in the cross-device setting. In the training and inference time, each client(device) needs to maintain $M$ samples and each sample has dim($z$), which may greatly increase the memory and computation cost if both $M$ and dim($z$) are large (compared to the algorithms that a client only needs to keep a single model locally) - this may limit the usecase of the proposed algorithm in resource-constrained clients, which is the case for edge devices such as mobile phones.

---

> > > ### Author Response · Authors · 2022-08-09
> > > **Response to your additional questions**
> > >
> > > Dear Reviewer 8AYd,
> > >
> > > Thank you for your additional comments which will bring more clarity to the revised version of the manuscript.
> > >
> > > To conduct this cross-device experiment in a stateless fashion, we indeed had to increase $M$ from 10 to 50, which, of course, required additional local computations; while at the same time decreasing the communication bottleneck.
> > >
> > > - For EMNIST, the size of $\phi$ is around 1M and is very close to what we used for CIFAR100 (except for the input of 28x28 images); the size of $z_i$ is roughly 8k, and the size of $\beta = (\mu,\sigma)$ is roughly 8k since we chose a Gaussian prior $N(\mu,\sigma^2I_d)$. To speed up the process in this experiment, we fixed the hyperparameter $\beta$ of the prior distribution.
> > >
> > > - For the Stack-Overflow dataset, we were inspired by the architecture of [1]. We performed next-word prediction using an RNN with a single LSTM layer. The network with shared parameters $\phi$ had 4M parameters and the fixe of the personalized parameters $z_i$ was 6.7M. Regarding $\beta = (\mu,\sigma)$, the size is roughly 6.7M and analogously to EMNIST, we fixed this hyperparameter to speed up convergence. Note that in this experiment, the size of $z_i$ is greater than the size of $\phi$. We did not have time to test other architectures that would better fit into our framework but this could be done in the revised version version of the manuscript if accepted.
> > >
> > > Regarding your comment on the applicability of the proposed approach in resource-constrained clients, we agree with you that the proposed algorithm is not a silver bullet and it (as long as other Bayesian FL methods) cannot be applied to use-cases featuring a very large neural network. Nevertheless, regarding our promising experimental results, we believe that the proposed framework can stand as a strong baseline for cross-device applications with a small/moderate personalized layer (size of $z_i$) requiring uncertainty quantification.
> > >
> > > These details, limitations and additional experiments will be added to the revised version of the manuscript if accepted.
> > >
> > > [1] Charles, Zachary, et al. "On large-cohort training for federated learning." Advances in neural information processing systems 34 (2021): 20461-20475.

---

> > > > ### Comment · Reviewer_8AYd · 2022-08-09
> > > > **Thanks for the additional information, and the discussions on the memory/computation limitations**
> > > >
> > > > Thanks for the additional information on the cross-devices experiments. It is a little surprising to see that on StackOverflow, the personalized parameters $z$ is even larger than $\phi$ (and since $M$ is 50, the proposed algorithm requires 50 times memory than other algorithms that only need a single model on each client), but I agree with you that other Bayesian FL methods have similar limitations. I also agree that the proposed algorithm can be a strong baseline when the model size is small (or when the client's local resources are not constrained, such as in the cross-silo setting). Given the promising experimental results, I will increase my score from 5 to 6.

---

### Official Review · Reviewer_BRBt · 2022-07-11

**Rating:** 5
**Confidence:** 3
**Soundness:** 3 good
**Presentation:** 2 fair
**Contribution:** 4 excellent

**Summary:**

This paper proposes a novel method for personalized FL having uncertainty quantification by using the population modeling paradigm where the client's models involve fixed common population parameters and random effects. From their empirical results, their method is robust to client drift, practical for inference, and uncertainty quantification under mild computational and memory overheads. Their method FedPop performs personalization and cheaply works for quantifying the uncertainty in an on-device manner.

**Questions:**

Q1) During the training dynamics, how and why can FedPop is robust to client drift, practical for inference on new clients?

Q2) Is your theorecial result empirically guaranteed after the FL training? how does the trained model behavior have?

Q3) Do your methods have robustness even in I.I.D. settings?

**Limitations:**

Despite their significant empirical results, there remain limitations: 1) More other methods should be conducted to verify its robustness in applications to advanced Personalized FL methods. 2) ablation studies on each factor of FedPop such as warm start strategy, gradient estimator etc... as well as hyperparamter settings of other regularization methods such as weight decay, augmentation...

**Strengths And Weaknesses:**

* Strength
- Their claim is supported by both theoretical and empirical results.
- The authors point out a major problem in related field.

* Weakness
-  the set-up of experiments and results reported in this paper are still questionable. (Datasets, Architectures, FL settings such as participation ratio, data heterogeneity settings with Dirichlet distribution)
- The empirical evidence for Theorem 1 may be needed.




------updated after rebuttal---------

Thank you for your detailed response for my raised concerns. still I think this work needs more comparisons with related FL approaches based on bayesian approaches, but most concerns are solved. So,  I decided to increase my score from 4 to 5.

---

> ### Author Response · Authors · 2022-08-02
> **Reponse to Reviewer BRBt (1/2)**
>
> We would like to thank the anonymous reviewer for the constructive feedback and relevant questions. Please find below our answer to all your questions and concerns.
>
> **1. During the training dynamics, how and why can FedPop be robust to client drift, and practical for inference on new clients?**
>
> We thank the reviewer for this important remark which highlights two benefits of the proposed methodology $\texttt{FedPop}$.
>
> **- Robustness to client drift.** For simplicity, we will take the example of $\texttt{FedSOUL}$ (see Algorithm 1) which uses the Markov kernel associated with Langevin Monte Carlo to compute gradient estimates of the local marginal likelihood.
>     However, our answer holds for general  Markov kernels (adjusted or unadjusted).
>     In this scenario, $M$ steps of Langevin Monte Carlo are performed on each device to draw samples $Z_k^{(i,m)}$ used to compute Monte Carlo estimates $I_k^{(i)}$ and $J_k^{(i)}$.
>     Increasing the number of local steps $M$ does not slow down convergence but instead allows for more accurate Monte Carlo integration and hence better convergence properties.
>     In contrast, the client drift phenomenon for classical FL approaches (e.g. $\texttt{FedAvg}$) slows down convergence as the number of local iterations $M$ increases.
>
> **- Simple inference on new clients.**
>     Typical personalized FL approaches such as $\texttt{DITTO}$ or $\texttt{FedRep}$ require additional local training for inference on new clients.
>     In contrast, the proposed $\texttt{FedPop}$ methodology allows for a cheaper two-step approach once we have estimated $\theta^\star = (\phi^\star,\beta^\star)$, as detailed below for a new client $b+1$ with feature vector $x$ :
>
> 1. Sample $(Z_{b+1}^{(k)})_{k \in [K]}  \sim p(\cdot \mid \beta^\star)$ in a i.i.d. manner.
>
> 2. Estimate the posterior predictive function by $p(\cdot \mid x) \approx K^{-1} \sum_{k=1}^K  p(\cdot \mid \phi^\star, Z_{b+1}^{(k)},x)$.
>
> The prior $p(z_{b+1} \mid \beta^\star)$ is typically chosen so that sampling is computationally cheap, e.g. a Gaussian with diagonal covariance matrix as in our experiments.
>
> We will add this discussion in the revised version of the manuscript.
>
> **2. Is your theoretical result empirically guaranteed after the FL training? how does the trained model behavior have?**
>
> We verified that our theoretical result is guaranteed empirically in the first experiment that is presented in our paper. In this illustration, we consider synthetic data with Gaussian distributions and matching our assumptions. To compare our algorithm with a non-FL setting, we perform a non-distributed and non-federated stochastic approximation algorithm to find $\theta^\star$ using a large number of iterations to get an accurate approximation of the optimal parameter $\theta^\star$.
> Then, we use $\texttt{FedPop}$ to obtain an estimate $\tilde{\theta}^\star$  and measure the relative error in $\ell_2$-distance between $\theta^\star$ and $\tilde{\theta}^\star$.
> For some outer iterations $T=100$, the relative error was less than $10^{-3}$, which illustrates the relevance of our theorems.
>
> We will add these additional results in the revised version of our manuscript.
>
> **3. Do your methods have robustness even in I.I.D. settings?**
>
> Our focus is on the non-i.i.d. setting where robustness is defined regarding data heterogeneity. In the i.i.d. setting, our theoretical results also hold naturally in that context by discarding the parameter $\beta$.
> Regarding empirical performances, we will add additional numerical simulations in the revised version of the manuscript under the i.i.d. scenario.
> We did not have the time to run these experiments during the rebuttal period since we ran other experiments to address your next concern.
>
>
> **4. More other methods should be conducted to verify its robustness in applications to advanced Personalized FL methods.**
>
> As pointed out in Table 2 in our paper, we already compared our approach with *nine* competing and state-of-the-art personalized FL methods. This already shows the robustness of our approach on the considered set of experiments.
> Nevertheless, we compared the robustness of our approach with competitors on additional datasets and observe the same conclusions regarding the performances of our approach.
> The results can be seen following the link: https://github.com/anonneurips1/rebuttal_neurips/tree/main.

---

> > ### Author Response · Authors · 2022-08-02
> > **Response to Reviewer BRBt (2/2)**
> >
> > **5. Ablation studies on each factor of FedPop such as warm start strategy, gradient estimator etc... as well as hyperparameter settings of other regularization methods such as weight decay, augmentation....**
> >
> > - **Gradient estimates.** We could have proposed other estimates based on control variates to reduce the impact of data heterogeneity on convergence but this would boil down to proposing other FL algorithms. As such, this analysis is out of the scope of this paper but would constitute an interesting follow-up.
> > - **Warm-start.** This analysis is very interesting and has also been proposed by Reviewer 8AYd. We ran several experiments to check if we are capable to reproduce our previous results, using a stateless (without warm-starts) version of our algorithm. Instead of starting from the previous state for a particular client, we sample a new initial state from the prior $p(Z|\beta)$ at each round. However, as expected, obtaining results similar to the warm-start scenario requires a larger number of inner iterations $M$ because there is now a transient phase at each iteration of the algorithm. Concretely, our new experiments show that $M$ has to be set to 50 in this case (in contrast to 10 in the warm-start scenario). We end up with an interesting trade-off: if client-server communication is a bottleneck, we can use a stateless version of the algorithm at the price of longer sampling procedures on each client. On the other hand, if local steps are expensive and the main limitation (for example when clients are mobile devices), we would prefer to use a stateful version of our algorithm and keep warm-start initializations.
> >
> > - **Hyperparameter settings.** The intuition of selecting particular optimizers is described in the next paragraph. Additionally, we tuned hyperparameters such as learning rate, using Optune package which performs Bayesian hyperparameter optimization.
> >
> >
> > **6. The set-up of experiments and results reported in this paper are still questionable.**
> >
> > In our experimental section, we were inspired by the experimental setup of the paper inducing the $\texttt{FedRep}$ method published at ICML 2021, and we precisely followed their setting regarding data partitioning and neural network architectures.
> >
> > - **Datasets.** Specifically, we synthetically generated heterogeneity of our datasets (CIFAR10 and CIFAR100) by distributing different numbers of classes to each client. If not stated otherwise, each client is assigned the same number of training samples (for these datasets it is 50000 / $b$, where $b$ is the number of clients). Following $\texttt{FedRep}$, the maximum number of data points for each client was restricted to 500 and we used the same data augmentation procedure. More precisely, for the training phase, we performed a random horizontal flip and normalization using statistics of the full train dataset. During the testing phase, no data augmentation was performed, but we still use the same normalization of the training phase.
> >
> > - **Model architectures.** Again, we followed the $\texttt{FedRep}$ paper here. Architecture depends on the dataset that we considered. For the fixed-effect parameters on CIFAR10, we used a convolutional neural network with two convolutional layers followed by two fully-connected layers with ReLU activation.
> >  For CIFAR100's fixed-effect parameters, we chose the same architecture but with larger numbers of units and channels.
> > The first fully-connected layer takes 3200 units and transforms them to 256, the second: 256$\rightarrow$128. In contrast to CIFAR10's shared model, we use a dropout layer (with the probability of an element to be zeroed at 0.6).
> > For random-effect parameters, we use the same architecture for both datasets: one fully-connected layer but with different input (64 for CIFAR10 and 128 for CIFAR100) and output (10 for CIFAR10 and 100 for CIFAR100) sizes.
> >
> > - **Training procedure.** All the methods (expect $\texttt{FedSOUL}$) were trained using the code of $\texttt{FedRep}$ following their complete implementation described in their Appendix.
> > For our model, we found, that it is slightly more beneficial to use the RMSprop optimizer for $\beta$ with a base learning rate of ~0.001.
> > For the fixed-effect parameter $\phi$, we used the Adam optimizer with a learning rate of 0.001. Parameters of the prior model (mean and standard deviation of Gaussian) were optimized with Adam optimizer as well with a learning rate of 0.1.
> > We also noticed that learning rate schedulers could give us some minor improvements (2-3\%). We used a cyclical learning rate scheduler for $\beta$ (typically with a gamma of 0.2 and exponential decay with stepsize 10) and a multistep LR scheduler for $\phi$ and prior parameters (with the gamma of 0.5 for prior / 0.7 for shared parameters and milestones at each 10th outer iteration for prior / 10th and 20th for shared parameters).

---

### Official Review · Reviewer_f916 · 2022-07-12

**Rating:** 6
**Confidence:** 3
**Soundness:** 3 good
**Presentation:** 3 good
**Contribution:** 3 good

**Summary:**

This paper introduces Bayesian paradigm to personalized FL, allowing for both personalization and cheap uncertainty quantification. Specifically, this paper introduces the fixed part (hyperparameter \phi) to capture the common representation of all local data and the random part via a common prior modeling data heterogeneity. The authors derive the non-asymptotic convergence guarantees for their federated stochastic optimization algorithm.

**Questions:**

Please see the weaknesses part.

**Limitations:**

As I can see, this paper has no potential negative societal impact.

**Strengths And Weaknesses:**

Merits:

1) The paper tackles an important problem in federated learning, which allows uncertainty quantification for personalized FL.
2) FedPop reduced the memory costs by introducing the mixed-effects instead of high-dimensional hyperparameter.
3) This paper provides convergence guarantees for the proposed algorithm.

Cons:
1) The novelties of proposed algorithm are unclear to me. It seems the difference between Bayesian FL and FedPop lies in the design of probabilistic hyperparameter. FedPop use two hyperparameters instead of a high-dimensional hyperparameter, including a common prior, to reduce storage costs.
2) The algorithm uses many independence assumptions, for example the independence on \phi and \beta at line 114 and the independence between local likelihood functions below line 117.
3) The authors may introduce more discussions on the computational complexities and communication burdens.

---

> ### Author Response · Authors · 2022-08-02
> **Response to Reviewer f916 (1/2)**
>
>
> We would like to thank the anonymous reviewer for the constructive and positive feedback. Please find below our answer to all your questions and concerns.
>
> **1. The novelties of proposed algorithm are unclear to me. It seems the difference between Bayesian FL and FedPop lies in the design of probabilistic hyperparameter. FedPop use two hyperparameters instead of a high-dimensional hyperparameter, including a common prior, to reduce storage costs.**
>
> We thank the reviewer for this important remark.
> We indeed agree that Section 4 $\textit{Related Works}$ in our paper mainly points out the storage benefits of $\texttt{FedPop}$ due to the use of two parameters, $\phi$ and $z_i$, instead of one high-dimensional parameter.
>
> In the revised version of the manuscript, we will further clarify the key differentiators of the proposed methodology $\texttt{FedPop}$ compared to existing Bayesian FL approaches.
> For the sake of completeness, these differentiating elements are described in what follows.
>
> **Novel FL framework based on mixed-effects modeling.** Bayesian FL methods aim at sampling a random parameter $\theta$ from a target probability distribution $\pi(\theta) \propto e^{-f(\theta)}$ where $f(\theta) = \frac{1}{b} \sum_{i=1}^b f_i(\theta)$ with $f_i$ denoting the negative log-likelihood associated to the $i$-th client.
>     In contrast, the proposed methodology $\texttt{FedPop}$ considers a mixed-effects modeling approach where parameters are divided into two categories: a fixed component $\phi$ and a random one $z_i$ for each client $i \in [b]$.
>     As such, the mixed-effects approach is in essence an empirical Bayesian/marginal likelihood approach (see [1, 2] below).
>     It corresponds to a hierarchical model that aims to combine the modeling flexibility and uncertainty assessment of Bayesian inference with computational pragmatism.
>     More precisely, a part of the parameters (fixed-effects $\phi$) are estimated via marginal likelihood maximization and the rest (random effects $z_i$) using common Bayesian techniques, which are in most cases low dimensional. As a result, up to our knowledge, the model and approach that we propose is novel in FL and comes with many benefits; see the next paragraph.
>
>
> **Novel inference algorithm based on stochastic approximation.** Since the parameter $\theta$ in Bayesian FL approaches is assumed to be random, Markov chain Monte Carlo (MCMC) and variational Bayes techniques have been proposed to sample from the corresponding posterior.
>     In contrast, since $\texttt{FedPop}$ involves both fixed and random parameters, another class of inference algorithms has to be considered.
>     More precisely, we proposed a novel class of *federated stochastic approximation algorithms* which involve (i) stochastic optimization to update the fixed parameter $\phi$, and (ii) MCMC to update the random part $z_i$.
>
> **Bridges the gap between optimization-based FL and Bayesian FL.**
>     The methodology based on mixed-effects modeling we are proposing also allows bridging the gap between *FL approaches based on optimization* and *Bayesian FL*.
>     Indeed, as emphasized at the end of Section 4 of the main paper, our framework reduces to $\texttt{FedRep}$ (optimization-based FL) when $\beta \rightarrow \infty$ and to $\texttt{DG-LMC}$ (Bayesian FL) when $z_i = \{\emptyset\}$.
>
> Note that this unifying property comes directly from our mixed-effects modeling approach and cannot be obtained via either optimization-based FL or Bayesian FL.

---

> > ### Author Response · Authors · 2022-08-02
> > **Response to Reviewer f916 (2/2)**
> >
> > **2. The algorithm uses many independence assumptions, for example, the independence on $\phi$ and $\beta$ at line 114 and the independence between local likelihood functions below line 117.**
> >
> > We thank the reviewer for this remark.
> > The first assumption has been made for the sake of simplicity and can be removed.
> > The second assumption is inherent to the FL setting we are considering and as such not restrictive.
> > Details supporting these claims are provided hereafter.
> >
> > 1. The assumption l.114 stating that $p(\phi,\beta) = p(\phi)p(\beta)$ has been made for the sake of simplicity and can be generalized easily. Indeed, we can set a general joint prior $p(\phi,\beta)$ (e.g. a joint Gaussian prior with a non-diagonal covariance matrix) and perform stochastic optimization via the update steps
> >
> >     $\beta_{k+1} = \Pi_{\mathsf{B}}\left(\beta_k+ \eta_{k+1}^{(1)} \left[\nabla_\beta\log p(\phi_k,\beta_k) + \sum_{i=1}^b g^{(i)}_k(\phi_k,\beta_k)\right]\right),$
> >
> >      $\phi_{k+1} = \Pi_{\Phi}\left(\phi_k + \eta_{k+1}^{(2)}\left[\nabla_\phi \log p(\phi_k,\beta_k) + \sum_{i=1}^b h^{(i)}_k(\phi_k,\beta_k)\right]\right).$
> >
> > 2. The assumption on local likelihood functions l.117 is not an assumption but a core property of the FL paradigm, see e.g. the review paper "Advances and Open Problems in Federated Learning" by Kairouz et al. (2021).
> >     Indeed, in federated learning, local datasets $(D_i)_{i=1}^b$ are assumed to be generated locally and independently on each device.
> >
> > These elements will be added to the revised version of the manuscript.
> >
> > **3. The authors may introduce more discussions on the computational complexities and communication burdens.**
> >
> > We agree with the reviewer. These discussions, detailed below, will be added to the revised version of the manuscript.
> >
> > 1. **Computation complexity.** Compared to standard FL methods, our approach has an additional computational cost on the client side associated with Monte Carlo approximations $I_k^{(i)}$ and $J_k^{(i)}$.
> >     In practice, this cost is negligible.
> >     Indeed, in our experiments, we found that using a small value of $M \in [1,10]$ was sufficient to obtain state-of-the-art results in terms of accuracy on the test dataset.
> >     We would like to emphasize that this additional computational cost has also two side advantages compared to existing FL approaches: (1) it allows us to communicate less frequently with the central server and (2) it allows us to converge faster when the number of local iterations $M$ increases since Monte Carlo approximation becomes better.
> >
> > 2. **Communication overhead.** As pointed out in Table 1 in the main paper, our methodology $\texttt{FedPop}$ improves upon existing FL approaches regarding the communication overhead.
> >     Indeed, $\texttt{FedPop}$ offers the flexibility to use both *compression* for sending updates to the server, and *multiple local steps* to reduce the communication frequency.
> >     As such, depending on the bandwidth and local computational power, the practitioner can adapt the number of local iterations $M$ and the parameter of the compression operator.
> >     Up to our knowledge, this work is the first one combining compression and multiple local steps for personalized FL.
> >
> > [1]  Robbins, Herbert (1956). "An Empirical Bayes Approach to Statistics".
> >
> > [2] Casella, George (1985). "An Introduction to Empirical Bayes Data Analysis"

---

> > > ### Comment · Reviewer_f916 · 2022-08-09
> > > **Thanks for your response**
> > >
> > > Your detailed response have solved my raised concerns. I have no further questions.

---

### Meta-Review · Area_Chair_HoyE · 2022-08-26

**Recommendation:** Accept
**Confidence:** Certain

**Metareview:**

The reviewers agree that the paper makes significant progress on uncertainty quantification for personalized federated learning. This is clearly a fundamental problem. The new approach is shown to have theoretical guarantees. The paper is easy to read. Based on the above, I recommend acceptance. Meanwhile, please carefully revise the paper and incorporate new experimental results in the final version according to the reviews.

**Award:**

No

---

### Decision · Program_Chairs · 2022-09-14

Accept